# Performance Improvement of NIR Spectral Pattern Recognition from Three Compensation Models’ Voting and Multi-Modal Fusion

**DOI:** 10.3390/molecules27144485

**Published:** 2022-07-13

**Authors:** Niangen Ye, Sheng Zhong, Zile Fang, Haijun Gao, Zhihua Du, Heng Chen, Lu Yuan, Tao Pan

**Affiliations:** Department of Optoelectronic Engineering, Jinan University, Guangzhou 510632, China; yng2020@stu2019.jnu.edu.cn (N.Y.); zhongsheng@stu2019.jnu.edu.cn (S.Z.); fangzile@stu2019.jnu.edu.cn (Z.F.); haijungao@stu2019.jnu.edu.cn (H.G.); dzh0905@stu2019.jnu.edu.cn (Z.D.); hengchen@stu2019.jnu.edu.cn (H.C.); yuanlu9835@stu2020.jnu.edu.cn (L.Y.)

**Keywords:** near-infrared spectroscopic pattern recognition, multi-optical-path, two-category priority’s compensation models, three-model voting fusion, moving-window-*k*-nearest neighbor, Norris derivative filter

## Abstract

Inspired by aquaphotomics, the optical path length of measurement was regarded as a perturbation factor. Near-infrared (NIR) spectroscopy with multi-measurement modals was applied to the discriminant analysis of three categories of drinking water. Moving window-*k* nearest neighbor (MW-kNN) and Norris derivative filter were used for modeling and optimization. Drawing on the idea of game theory, the strategy for two-category priority compensation and three-model voting with multi-modal fusion was proposed. Moving window correlation coefficient (MWCC), inter-category and intra-category MWCC spectra, and *k*-shortest distances plotting with MW-kNN were proposed to evaluate weak differences between two spectral populations. For three measurement modals (1 mm, 4 mm, and 10 mm), the optimal MW-kNN models, and two-category priority compensation models were determined. The joint models for three compensation models’ voting were established. Comprehensive discrimination effects of joint models were better than their sub-models; multi-modal fusion was better than single-modal fusion. The best joint model was the dual-modal fusion of compensation models of one- and two-category priority (1 mm), one- and three-category priority (10 mm), and two- and three-category priority (1 mm), validation’s total recognition accuracy rate reached 95.5%. It fused long-wave models (1 mm, containing 1450 nm) and short-wave models (10 mm, containing 974 nm). The results showed that compensation models’ voting and multi-modal fusion can effectively improve the performance of NIR spectral pattern recognition.

## 1. Introduction

The water system samples, (e.g., drinking water, dairy product, and blood) are important analysis objects in the fields of food, environment, and biomedicine. Water can be used not only as a direct analysis object, (e.g., the safety detection and classification identification of drinking water) but can also be used as the most common background solvent of liquid samples, which is a very important analyte. Qualified drinking water has undergone strict safety testing before it can enter the market for social use. Due to the huge market demand for drinking water, some brands of high-end bottled drinking water are expensive and increasingly popular, and they can be easily counterfeited. Using water that has not undergone safety testing to counterfeit high-end drinking water will not only damage the rights and interests of producers and consumers but may also cause large-scale safety problems of drinking water. The authenticity identification of drinking water brands on the market is an important problem that needs to be solved urgently. The current water quality testing methods mainly include quantitative analysis of water quality safety indexes (multiple trace components), which are complex and expensive. Due to the very similar component structures, the above quantitative methods are still difficult to achieve accurate identification of different drinking water brands, and there have been no related results reported so far. Therefore, a fast and easy detection method that can be used in the field is urgently needed.

Near-infrared (NIR) spectra mainly reflect the vibrational absorption of the overtones and combined frequencies of the hydrogen-containing groups X-H. This measurement method usually does not require reagents to measure samples directly, which has the advantage of being quick and easy. Water system samples have significant NIR absorption, and NIR spectroscopy has been applied to a variety of water-based analysis samples in agriculture and food [1,2,3,4,5], environment [6,7], and biomedicine [8,9,10,11,12].

The qualitative discriminant analysis of NIR spectroscopy is one of the hot research directions in recent years. For the classification and identification of samples with small differences in component content, the discriminant analysis is more effective and simpler than quantitative analysis. This discriminant analysis involves pattern recognition technology of multiple spectral populations, which needs to make full use of the spectral similarities of the same population, and spectral differences of different populations. It has been applied to sample identification in many fields in recent years, such as melon genotype [13], edible oil type [14], transgenic sugarcane leaf [15,16], milk powder adulteration identification [17], wine identification [18], rice seed authenticity identification [19] and thalassemia screening [16,20]. However, the application of NIR spectral discriminant analysis to the identification of water samples has not been reported yet. Efficient discriminant analysis methods for spectral populations with small differences are also rare.

Water sample identification involves multi-category discriminant analysis of spectral populations, which is more challenging than the two-category discriminant analysis problems. Partial least squares discriminant analysis (PLS-DA) is a well-performed method of two-category discriminant analysis. When using PLS-DA for multi-category discriminant analysis, it is necessary to perform multiple two-category analysis and their comprehensive evaluation. This process is complicated and difficult to popularize. Principal component analysis–linear discriminant analysis (PCA–LDA) is another effective two-category discriminant analysis method. When using PCA–LDA to process *n*-category discriminant analysis, it is necessary to determine the optimal classification surface of *n*-1 dimensions in *n*-dimensional space, which is mathematically complicated and difficult. Therefore, PLS-DA and PCA–LDA methods are not suitable to deal with the multi-classification problem.

The *k*-nearest neighbor (kNN) [21,22,23,24,25] is one of the most commonly used multi-classification algorithms. When it was applied to a supervised multi-category problem, the idea was as follows: based on the spectra of calibration samples containing multiple categories, the Euclidean (or Mahalanobis) distances between the unknown sample and all calibration samples were calculated; the *k* nearest calibration samples were determined; finally, the unknown sample was categorized as the category with the largest number among the *k* nearest samples.

The kNN uses two cyclic parameters of sample number and *k* value, and determines the optimal *k* based on discriminant effect, which has better robustness and applicability than the ordinary Euclidean distance classification method. It is not limited by the number of categories and is especially suitable for multi-category spectral discriminant analysis. The kNN has been applied to multi-category discriminant analysis based on various spectral techniques, such as, NIR [21], mid-infrared [22], Raman [23,24], and laser-induced breakdown spectroscopies [25].

On the other hand, the wavelength model optimization can enhance the characteristic attributes of the spectral category, reduce the interference of redundant data and model complexity, and provide valuable reference for the design of a dedicated spectrometer. Since further wavelength optimization requires higher-dimensional algorithm integration, there are few works on kNN-based wavelength model optimization.

The moving-window waveband screening combined with PLS regression [2,3,9,10] is a well-executed method in the quantitative analysis of NIR spectroscopy. However, its combination with qualitative discriminant analysis algorithms, (e.g., kNN) is still rare. In the current study, “tap water” and two kinds of drinking water brands (C’estbon and Nongfu Spring), a total of three categories of water samples, were used as identification samples. NIR spectroscopy combined with kNN were used to establish the discriminant analysis models for three categories of drinking water. An ensemble algorithm based on kNN and moving-window waveband screening was established, denoted as MW-kNN, and applied to the wavelength optimization of the three-category discrimination model for drinking water. Among them, the initial wavelength and the number of wavelengths were used as the cyclic parameters of MW-kNN to realize the modeling and optimization of all sub-wavebands.

Considering the small spectral differences of different categories of water, new in-depth research on population differences in water spectra is required. The recently emerging aquaphotomics method [26,27,28] uses specific disturbance factors, (e.g., temperature, pH) and makes changes to the dynamic measurement of water-system samples to obtain a “curved surface” spectrum of water absorption peak disturbances, that is, the spectral disturbance set of a sample, and further use omics method to achieve quantitative and qualitative analysis of weak features. The curved surface spectrum has rich information features with higher dimensions than the line spectrum, which can enhance the information capacity or population difference of the samples, thereby improving the accuracy of quantitative and qualitative analysis of weak analytes. The water spectrum has one strong absorption peak near 1940 nm in the NIR combination frequency region (1900–2500 nm), twelve slightly weaker absorption peaks in NIR double overtones frequency region (1300–1600 nm), and two weaker absorption peaks in the NIR high-overtone frequency region (900–1300 nm) [26,27,28]. 

As we know, using a transmission measurement accessory with a longer optical path (>1 mm) can increase the significance of the water absorption peaks in the short-wave NIR region, while using a transmission measurement accessory with a shorter optical path (<1 mm) can avoid saturated absorption of water in long-wave NIR region. Therefore, using long and short optical path transmission accessories to extract the spectral information in the short- and long-wave NIR regions, respectively, and perform model fusion, is expected to comprehensively improve the differences of water spectral populations and improve the accuracy of spectral identification. Inspired by aquaphotomics, the optical path length of the transmission measurement accessory was used as a perturbation factor, and a novel method for multi-modal spectra fusion modeling based on multi-optical-path measurement was proposed. The multi-modal NIR spectra based on the short, medium, and long optical paths (1 mm, 4 mm, and 10 mm) were used to establish the discriminant analysis models for three categories of drinking water. 

A comprehensive judgment method based on the three models’ voting was also proposed for the model fusion. Each sample was judged three times, and the sample category of comprehensive judgment was obtained according to the principle of “two wins in three games”, and thereby a joint model of three-model voting was established. On the other hand, drawing on the idea of game theory, for the 3-category discriminant analysis problem, a strategy of two-category priority compensation was proposed and used for three-model voting fusion. Its goal was that the comprehensive discrimination effect of the fusion model was better than its three sub-judgments. Moreover, in the process of two-category priority compensation and three-model voting fusion, all joint models based on the same and different measurement modals were used. The joint voting of the multi-modal compensation models can highlight the spectral differences from two dimensions, which is expected to improve the discrimination accuracy of the spectral population with small differences. 

The strategy of three compensation models’ voting and multi-modal fusion was applied for performance improvement of NIR spectral pattern recognition of three categories of drinking water. Furthermore, the spectra of the moving-window correlation coefficient, intra-class correlation coefficient, and inter-class correlation coefficient were also proposed to evaluate weak differences between two spectral populations.

## 2. Results and Discussion

### 2.1. Spectra of Three Groups of Water Samples Based on Three Measurement Modals

Using the transmission accessories of 1, 4, and 10 mm cuvettes, the Vis–NIR spectra of three categories of water samples (one-, two-, three-category) in the desaturated wavebands are shown in Figure 1. In the case of the long optical path measurement, the spectra of the water samples exhibited saturated absorption, so the spectra of 4 and 10 mm cuvettes showed only the unsaturated region. In order to facilitate viewing in the same figure, the absorbance values of samples of two-, and three-category were increased by 1.5 and 3, respectively. Seeing Figure 1, the spectra in desaturated wavebands of three groups of water samples based on three measurement modals, no significant difference was observed.

Note that, in the spectra of the 1 mm measurement modal, one strong absorption peak near 1948 nm in the NIR combination frequency region, and one slightly weaker absorption peak near 1450 nm in the NIR double-overtone frequency region were observed; in the spectra of the 10 mm measurement modal, two weaker absorption peaks (974 nm, 1196 nm) in NIR high-overtones frequency region were observed, while in the spectra of the 4 mm measurement modal, the absorption peak near 1450 nm began to appear saturable absorption, the two weaker absorption peaks (974 nm, 1196 nm) were less obvious.

### 2.2. Intra-Category and Inter-Category Correlation Coefficient Spectra

In order to evaluate the weak difference between two spectral populations, the moving-window correlation coefficient (MWCC) spectrum between any two spectra was first proposed. In the symmetrical wavelength window (number of wavelengths: *m* = *p* + 1 + *p*), the correlation coefficient between the two spectra was calculated as the local correlation coefficient of the center wavelength. Through the moving window, the local correlation coefficients of each central wavelength were calculated, and thereby the MWCC spectrum corresponded to *m* was obtained, as follows:(1)$$R_{i,m}=\frac{\sum_{k=-\left(m-1\right)/2}^{\left(m-1\right)/2}\left(x_{i+k}-{\bar{x}}_{i,m}\right)\left(y_{i+k}-{\bar{y}}_{i,m}\right)}{\sqrt{\sum_{k=-\left(m-1\right)/2}^{\left(m-1\right)/2}{\left(x_{i+k}-{\bar{x}}_{i,m}\right)}^{2}\sum_{k=-\left(m-1\right)/2}^{\left(m-1\right)/2}{\left(y_{i+k}-{\bar{y}}_{i,m}\right)}^{2}}}$$
where *i* was the serial No. of central wavelength.

Next, for the two spectral populations, their respective average spectrum was calculated separately. Within each spectral population, the MWCC spectrum between each spectrum and the average spectrum was calculated, and then the average spectrum of all MWCC spectra was further calculated, called the intra-category MWCC spectrum. For two spectral populations, the MWCC spectrum between each spectrum in a spectral population and the average spectrum of another spectral population was calculated, and then the average spectrum of all MWCC spectra was further calculated, called inter-category MWCC spectrum.

Here, the MWCC spectrum, intra- and inter-category MWCC spectra of spectral populations of any two categories of samples were calculated to evaluate the weak differences between the two categories of water spectra. Since the numerical difference was small, the different spectrum of the intra- and inter-category MWCC spectra were further calculated to observe their differences.

For the spectral datasets of 1, 4, and 10 mm, the intra-category and inter-category MWCC spectra were calculated for the 1–3 category, 2–3 category and 1–2 category spectral groups, respectively. Additionally, the different spectra between two intra-category MWCC spectra and one inter-category MWCC spectrum were further calculated, see Figure 2. As can be seen from Figure 2a, for the dataset of the 1 mm measurement modal, the differences between the two categories of water spectra were mainly at 1072 nm, 1106 nm, 1198 nm, 1258 nm, 1448 nm, 1682 nm, 1806 nm, 1944 nm, 1948 nm, 2212 nm. Among them, 1448 nm was in the wavelength range of 1448–1454 nm (OH-(H_2_O)_4,5_, one of the twelve characteristic water wavelengths) [28], and 1944 nm, 1948 nm located near the strong absorption peak of water at 1948 nm (see Figure 1). As can be seen from Figure 2b, for the dataset of the 4 mm measurement modal, the differences between the two categories of water spectra were mainly at 798 nm, 858 nm, 980 nm, 1078 nm, 1104 nm, 1198 nm, 1258 nm, 1440 nm, 1454 nm. Among them, 1440 nm and 1454 nm were in the wavelength ranges of 1432–1444 nm and 1448–1454 nm, respectively (S_1_ and OH-(H_2_O)_4,5_, two of the twelve characteristic water wavelengths) [28], and they were also located near the absorption peak of water at 1450 nm (see Figure 1). As can be seen from Figure 2c, for the dataset of the 10 mm measurement modal, the differences between the two categories of water spectra were mainly at 794 nm, 980 nm, 1078 nm, 1196 nm, and 1260 nm. Among them, 980 nm and 1196 nm were located near the weaker absorption peaks of water at 974 nm and 1196 nm, respectively (see Figure 1). In the six difference spectra, positive values were almost all observed, indicating that the intra-category correlations were significantly better than inter-category correlations, thus indicating the existence of weak differences between the two categories of spectral populations.

### 2.3. MW-kNN Models

First, for the three modal spectral datasets, using direct kNN method, the three-category discriminant analysis models based on desaturated wavebands were established, respectively. See Table 1 for their optimal *k* values and modeling discriminant effects. The results showed that the case of 10 mm achieved a better discrimination effect (RAR_Total_ = 82.9%).

Then, the MW-kNN models were built for each modal dataset. For the two-dimensional parameter combination (*I*, *N*) of the initial wavelength (*I*) and the number of wavelengths (*N*), the 3D effect diagrams of the modeling discriminant effect (RAR_Total_) of kNN models of three measurement modals are shown in Figure 3. Among them, the indicated points of the arrow were the optimal parameter combinations (*I*, *N*). Additionally, the corresponding optimal wavebands of the three measurement modals (1 mm, 4 mm, 10 mm) were 966–1894, 938–1402, and 964–1378 nm, respectively.

The modeling discrimination effects of the optimal MW-kNN models of three measurement modals are summarized in Table 2. The results show that the modeling effect of the optimal MW-kNN models of the three modals were significantly better than the previous direct kNN models (Table 1), and the number of wavelengths used were also significantly reduced, so the models were simpler.

As described above, the optimal MW-kNN models reached the better modeling discrimination effects for three categories of water samples. As described in Figure 1, the spectral shapes of the three categories of water samples were very similar and no significant differences were observed. Therefore, it is necessary to reanalyze the differences of the three spectral populations based on the principle of the kNN method.

Note that the optimal *k* value for all three optimal MW-kNN models was 3 (Table 2). According to the principle of kNN, each prediction sample was determined as the category with the most occurrence among the categories of the three calibration samples that had the minimum distance from it. In view of this, based on the Euclidean distance of the spectra, a visualization method described the differences of spectral populations was proposed: for a specific spectral range, the distances of the spectrum of each prediction sample from the spectra of all calibration samples were calculated, and the first three shortest distances were selected and plotted corresponding to the serial number of the sample. Among them, if the selected distance was the distance between samples of the same category, its value was displayed in a green hollow circle, and if the selected distance was the distance between different samples, its value was displayed in a red hollow circle. Among the first three shortest distances of each prediction sample, if the number of green hollow circles was greater than or equal to 2, it indicated that the sample was correctly identified; otherwise, the sample was incorrectly identified.

The effect of the above visualization of spectral population differences was directly related to the selection of spectral waveband. Using the MW-kNN method for waveband selection can highlight the differences in spectral populations. Here, based on the optimal MW-kNN wavebands for three measurement modals, the visualization of spectral population differences for three categories of water samples was presented. To avoid crowding of data points, the first 60 prediction samples of each spectral population were used for display; the three points on the same horizontal coordinate represented the first three shortest distances of the same sample, which were connected by vertical lines for easy observation; in addition, a pentagram (red or green) was added at the bottom of the same horizontal coordinate to show whether the judgment was correct or not: green meant correct judgment, red meant wrong judgment. In the case of 1 mm, based on the optimal waveband (966–1894 nm), the first three shortest distances between the first 60 one-category prediction samples and all calibration samples are shown in Figure 4a. In the case of 4 mm, based on the optimal waveband (938–1402 nm), the first three shortest distances between the first 60 two-category prediction samples and all calibration samples are shown in Figure 4b. In the case of 10 mm, based on the optimal waveband (964–1378 nm), the first three shortest distances between the first 60 three-category prediction samples and all calibration samples are shown in Figure 4c. As can be seen from the figures, for those prediction samples, the first three shortest distances were mostly green, a small part was red, and the green pentagrams were much more than the red pentagrams, indicating that most samples were correctly identified. In particular, in the case of 10 mm, in almost all the three-category prediction samples, the corresponding first three shortest distances were green (green pentagrams), indicating that the three-category samples (10 mm) were precisely recognized (RAR_3_ = 99.3%).

In fact, based on the principle of kNN and the wavelength model optimization, a plotting approach of the first *k*-shortest distances was proposed here. It can describe the differences of spectral populations, and it is also expected to be applied to other analytic objects.

### 2.4. Two Categories Priority’s Compensation Models

Using the method of priority compensation model described later in Section 3.5, the two-category priority compensation models Φ_1,2_, Φ_1,3_, Φ_2,3_ of three modals were determined, respectively. The selected corresponding wavebands were 1704–1868, 858–1896, and 844–1890 nm for 1 mm, 798–1412, 986–1402, and 1080–1332 nm for 4 mm, and 970–1378, 960–1140, and 924–1384 nm for 10 mm. The modeling discrimination effects of the two-category priority compensation models of three measurement modals are summarized in Table 3.

### 2.5. Optimal Norris Parameters

Using the Norris derivative filter algorithm, based on each modal, the parameter optimization of spectral preprocessing for the optimal MW-kNN model and the two-category priority compensation models was performed.

Take the optimal MW-kNN model (waveband) of 1 mm modal as an example, for the two-dimensional parameter combination (*S*, *G*) of the number of smoothing points (*S*) and the number of differential gaps (*G*), the 3D-effect diagram (RAR_Total_) of all kNN models processed by NDF (*D* = 1) is shown in Figure 5. Additionally, the indicated point of the arrow was the optimal parameter combination (*S*, *G*).

The modeling discrimination effects of the optimal MW-kNN models processed by optimal NDF parameters are summarized in Table 4. Comparing Table 2 and Table 4, after using NDF preprocessing, the effect of each model had been significantly improved.

The modeling discrimination effect of the two-category priority compensation models of three measurement modals processed by optimal NDF parameters are summarized in Table 5. Comparing Table 3 and Table 5, after using NDF preprocessing, the performance of almost every model also had been significantly improved.

### 2.6. Joint Models Based on Three-Model Voting Fusion

#### 2.6.1. Fusion Based on Single-Modal Models

After processing by the optimal NDF parameters, using the method described later in Section 3.5, based on the three two-category priority compensation models of each single-modal, the joint model based on three-model voting was established. The modeling discrimination effect of the three joint models are summarized in Table 6. Comparing Table 4 and Table 6, for each measurement modal, the comprehensive effect of the joint model was better than that of the corresponding optimal MW-kNN model. Among them, the joint model of the 1 mm’s modal was the best.

#### 2.6.2. Fusion Based on Multi-Modal Models

The experimental design of three-model voting fusion based on multi-modal spectra was considered using the method described later in Section 3.5.

Firstly, various cases of the joint voting of three models based on two modals were considered: the three sub-models of priority compensation were first sorted as the basic order of Φ_1,2_, Φ_1,3_, Φ_2,3_, and then all possible structures and arrangements of two modals were filled in, such as 1 mm-Φ_1,2_,10 mm-Φ_1,3_, 1 mm-Φ_2,3_. They were divided into three groups (structures): 1 mm and 4 mm, 1 mm and 10 mm, 4 mm and 10 mm; each group (take 1 mm and 10 mm as an example) was further divided into two sub-groups, (e.g., two 1 mm and one 10 mm or two 10 mm and one 1 mm); according to the order of collocation, each sub-group had three cases in total, (e.g., the first sub-group included three cases of 1 mm-Φ_1,2_, 1 mm-Φ_1,3_, 10 mm-Φ_2,3_; 1 mm-Φ_1,2_, 10 mm-Φ_1,3_, 1 mm-Φ_2,3_, and 10 mm-Φ_1,2_, 1 mm-Φ_1,3_, 1 mm-Φ_2,3_). In summary, there were eighteen dual-modal fusion cases. Modeling had been completed for each case, and the optimal case is summarized in Table 7.

Secondly, various cases of the joint voting of three models based on three modals were considered: the three sub-models of priority compensation were still sorted as the basic order of Φ_1,2_, Φ_1,3_, Φ_2,3_, and then all possible structures and arrangements of three modals were filled in; that is, the three sub-models used for joint voting belong to three different modals (1mm, 4mm, 10mm), and belong to three different priority compensation types (Φ_1,2_, Φ_1,3_, Φ_2,3_), respectively, such as 1 mm-Φ_1,2_, 10 mm-Φ_1,3_, 4 mm-Φ_2,3_; the three modals were arranged sequentially, and there were six cases as follows: (1) 1 mm-Φ_1,2_, 4 mm-Φ_1,3_, 10 mm-Φ_2,3_; (2) 1 mm-Φ_1,2_, 10 mm-Φ_1,3_, 4 mm-Φ_2,3_; (3) 4 mm-Φ_1,2_, 1 mm-Φ_1,3_, 10 mm-Φ_2,3_; (4) 4 mm-Φ_1,2_, 10 mm-Φ_1,3_, 1 mm-Φ_2,3_; (5) 10 mm-Φ_1,2_, 1 mm-Φ_1,3_, 4 mm-Φ_2,3_; (6) 10 mm-Φ_1,2_, 4 mm-Φ_1,3_, 1 mm-Φ_2,3_. In addition, the global optimal MW-kNN models of the three modals were also used as sub-models for joint voting, that was, the fusion of 1 mm-Φ_global_, 4 mm-Φ_global_ and 10 mm-Φ_global_. In summary, there were seven different triple-modal fusion cases. Modeling had been completed for each case. The modeling discrimination effect of the optimal model of six joint models based on compensation models’ fusion is summarized in Table 7. The modeling discrimination effect of the joint model based on the global optimal MW-kNN models’ fusion is also summarized in Table 7.

Seeing Table 7, the three selected fusion models all reached high discrimination accuracy and good category balance, the optimal RAR_Total_ and RAR_SD_ are 96.0% and 2.4%.

#### 2.6.3. Comparison before and after Model Fusion

Table 8 showed the comparison of the modeling discrimination effects for the optimal joint models of the three-model voting fusion and their sub-models. It showed that the effect of each joint model was better than its sub-models. Among them, the highest increasement of discrimination accuracy rate was 22.0%.

### 2.7. Independent Validation

The validation samples that were not involved in modeling were used to validate the four optimal joint models of three-model voting fusion. Table 9 showed the validation discrimination effect of the four optimal joint models based on the fusion of single-modal or multi-modal models. The four joint models had reached high validation discrimination accuracy and good balance, the best of which was the dual-modal fusion of 1 mm-Φ_1,2_, 10 mm-Φ_1,3_, 1 mm-Φ_2,3_, the RAR_Total_ was 95.5%, and the corresponding waveband combination (nm) was 1704–1868 (1 mm), 960–1140 (10 mm) and 844–1890 (1 mm). Among them, two models in the NIR overtones frequency region of the 1 mm modal (containing the absorption peak at 1450 nm) and one model in the NIR high-overtone frequency region of the 10 mm modal (containing the absorption peak at 974 nm) were jointly used. 

In fact, if the optical-path length of the transmission measurement accessory is regarded as a perturbation factor, the three-model voting fusion modeling approach based on multi-measurement modals is similar to the idea of aquaphotomics. The comprehensive use of the multi-modal spectral features of samples can improve the accuracy of discrimination.

Figure 6 showed the schematic diagrams of the recognition effect of the optimal joint model (dual-modal) based on a three-model voting fusion for the validation samples. Among them, the true value of the category value of the *i*-th sample was set to *i*, *i* = 1, 2, 3; its predicted value was set to $\delta_{i}$, $\delta_{i}\in$ {1,2,3}; when $\delta_{i}\;$= *i*, the recognition was correct, otherwise was wrong. The second category of drinking water samples reached the highest validation discrimination accuracy rate (RAR_2_ = 98.1%).

## 3. Materials and Methods

### 3.1. Experimental Materials, Instruments, and Measurement Methods

Two kinds of drinking water brands (C’estbon and Nongfu Spring) and “tap water” for users after municipal treatment, a total of three categories of water samples were collected as identification samples (not in order, denoted as 1-category, 2-category, 3-category). Among them, 162 tap water samples were collected successively from multiple tap water supply points on the university campus where the experiment was located; C’estbon and Nongfu Spring were purchased from regular commercial channels, 162 bottles of each category, one sample was taken from each bottle; in total, 486 samples were obtained. Two drinking water brand samples were purchased through formal channels, so the authenticity of their brands can be guaranteed, and their sample categories can serve as a reference for spectral pattern recognition.

The XDS Rapid Content^TM^ Liquid Grating Spectrometer (FOSS, Denmark) and a transmission accessory with multiple cuvettes were used for spectral measurement. Spectral scope ranged from 780–2498 nm with a 2 nm wavelength interval. Wavebands of 780–1100 nm and 1100–2498 nm were used for Si and PbS detection, respectively. Using the cuvettes of three optical paths of short, medium, and long (1 mm, 4 mm, 10 mm), each sample was measured three times. The obtained NIR spectra of multiple measurement modals were used for modeling and validation. The experimental temperature and humidity were 25 ± 1 °C and 45 ± 1%, respectively.

### 3.2. Calibration-Prediction-Validation Framework and Evaluation Indicators

A sample-independent experimental design in calibration–prediction–validation was adopted. The spectral data of the calibration and prediction sets were used for modeling and parameter optimization; the spectral data of the validation set that does not participate in the modeling was used to validate the established models. For each measurement modal, each category of water samples (162) was randomly divided into the calibration (60), prediction (50), and validation (52) sets. In total, for the numbers of the samples and spectra, the division was calibration (180 samples, 540 spectra), prediction (150 samples, 450 spectra), and validation sets (156 samples, 468 spectra). 

The evaluation indicators were set as the recognition accuracy rate (RAR*_i_*, *i* = 1, 2, 3) of each category sample and their standard deviation (RAR_SD_), as well as the total recognition accuracy rate (RAR_Total_) of all samples, as follows:(2)$${\mathrm{RAR}}_{i}=\frac{\tilde{M_{i}}}{M_{i}},\;i=1,2,3$$
(3)$${\mathrm{RAR}}_{\mathrm{Total}}=\frac{\overset{3}{\underset{i=1}{\sum }}\tilde{M_{i}}}{\overset{3}{\underset{i=1}{\sum }}M_{i}}$$
where $M_{i}\left(i=1,2,3\right)$ was the number of samples of *i*-th category of the prediction (or validation) set, and $\tilde{M_{i}}$ was the number of accurately identified samples in *i*-th category samples of the prediction (or validation) set. In the modeling process, the global optimal model was preferred according to the indicator RAR_Total_. In order to consider balance, RAR_SD_ was used as the second optimization indicator.

### 3.3. Norris Derivative Filter Algorithm

In spectral preprocessing, appropriate smoothing and derivatives can effectively eliminate noises and improve spectral information quality. The famous Norris derivative filter (NDF) is an effective spectral pretreatment method, which is an algorithm group with various parameters [29,30]. NDF includes two steps: the moving average smoothing and differential derivation and uses three parameters: the derivative order (*D*), the number of smoothing points (*S*, odd), and the number of differential gaps (*G*) [12,31,32]. Here, the loop parameters were set to *D* = 0, 1, 2; *S* = 1, 3…, 49; *G* = 1, 2…, 30. Any combination of parameters (*D*, *S*, *G*) corresponding to the Norris derivative mode, a total of 25 + 2 × 25 × 30 = 1525 modes were obtained. Based on variation of multiple parameters, the Norris derivative spectra were more diverse and applicable than the raw spectra. Norris parameters combination (*D*, *S*, *G*) should not be specified artificially but should be reasonably optimized according to the analysis object and modeling discrimination effect. Next, an algorithm platform for Norris parameter optimization based on kNN modeling was constructed by using MATLAB version 7.6 software.

For all Norris derivative modes, using the kNN algorithm, the calibration–prediction models were established corresponding to the Norris derivative spectra, which were called the Norris-kNN models. The recognition accuracy rate (RAR*_i_*, *i* = 1, 2, 3) of each category of prediction samples and the total recognition accuracy rate (RAR_Total_) of all prediction samples were calculated. Additionally, based on the total prediction effect (RAR_Total_), the optimal Norris parameters combination (*D*^*^, *S*^*^, *G*^*^) was preferred, as follows:(4)$$\mathrm{RA}{R\;}_{\mathrm{Total}}\;\left(D^{*},S^{*},G^{*}\right)=\overset{}{\underset{\begin{array}{l} D\in \{0,1,2\}\\ S\in \{1,3,\cdots ,49\}\\ G\in \{1,2,\cdots ,30\} \end{array}}{\mathrm{Max}}}\mathrm{RA}{R\;}_{\mathrm{Total}}\;(D,S,G).$$

### 3.4. MW-kNN Algorithm

Moving-window waveband screening and kNN were combined. The total indicator RAR_Total_ was adopted as the optimization goal and taken into account the balance indicator RAR_SD_, and a wavelength optimization platform (MW-kNN) of qualitative discriminant analysis was established by using MATLAB version 7.6 software. The parameters of MW-kNN were set as: (a) the initial wavelength *I*, (b) the number of wavelengths *N*. By looping these two parameters, all sub-wavebands were traversed to establish all kNN models, and the optimal model was determined based on the total prediction effect (RAR_Total_). 

Corresponding to the spectral unsaturated absorption regions of the three measurement models, the parameters of moving-window waveband screening were set as the follows: for the dataset of 1 mm, *I*∈{780, 782, ⋯, 2498}, *N*∈{2, 3, ⋯, 859, 860}; for the dataset of 4 mm, *I*∈{780, 782, ⋯, 1880}, *N*∈{2, 3, ⋯, 550, 551}; for the dataset of 10 mm, *I*∈{780, 782, ⋯, 1388}, *N*∈{2, 3, ⋯, 304, 305}. The parameter *k* of kNN was set as *k*∈{1, 2, ⋯, 5}. For the three datasets of 1 mm, 4 mm and 10 mm, based on the total prediction effect (RAR_Total_), the optimal parameters combinations (*I*^*^, *N*^*^, *k*^*^) were preferred, respectively, as follows: (5)$$\mathrm{RA}{R\;}_{\mathrm{Total}}\;(I^{*},N^{*},k^{*})=\overset{}{\underset{\begin{array}{l} I\in \{780,782,\cdots ,2498\}\\ N\in \{2,3,\cdots ,859,860\}\\ k\in \{1,2,\cdots ,5\} \end{array}}{\mathrm{Max}}}\mathrm{RA}{R\;}_{\mathrm{Total}}\;(I,N,k),$$
(6)$$\mathrm{RA}{R\;}_{\mathrm{Total}}\;(I^{*},N^{*},k^{*})=\overset{}{\underset{\begin{array}{l} I\in \{780,782,\cdots ,1882\}\\ N\in \{2,3,\cdots ,550,551\}\\ k\in \{1,2,\cdots ,5\} \end{array}}{\mathrm{Max}}}\mathrm{RA}{R\;}_{\mathrm{Total}}\;(I,N,k),$$
(7)$$\mathrm{RA}{R\;}_{\mathrm{Total}}\;(I^{*},N^{*},k^{*})=\overset{}{\underset{\begin{array}{l} I\in \{780,782,\cdots ,1388\}\\ N\in \{2,3,\cdots ,304,305\}\\ k\in \{1,2,\cdots ,5\} \end{array}}{\mathrm{Max}}}\mathrm{RA}{R\;}_{\mathrm{Total}}\;(I,N,k).$$

### 3.5. Strategy for 2-Category Priority’s Compensation and Three-Model Voting Fusion

Drawing on the idea of game theory, the strategy for 2-category priority’s compensation and three-model voting fusion was proposed.

The two-category priority compensation models were proposed first. For any two categories of samples of *i*, *j* (*i*, *j* = 1, 2, 3, *i*
< *j*), the total recognition accuracy rate (RAR*_i_*_, *j*_), as follows:(8)$${\mathrm{RAR}}_{i,j}=\frac{\tilde{M_{i}}+\tilde{M_{j}}}{M_{i}+M_{j}},$$
where *M_i_*, *M_j_* were the number of samples of *i*-th category and *j*-th category of the prediction (or validation) set, respectively, and $\tilde{M_{i}}$, $\tilde{M_{j}}$ were the number of accurately identified samples in *i*-th category and *j*-th category samples of the prediction (or validation) set, respectively. For the three categories of samples of 1, 2, and 3, there were three compensation models for the two categories of priority: Φ_1,2_, Φ_1,3_, Φ_2,3_. Among them, according to the optimal total recognition accuracy rate (RAR*_i_*_, *j*_) of *i*, *j*-category samples, the corresponding compensation model (Φ*_i_*_, *j*_) was selected, *i*, *j* = 1, 2, 3, *i*
< *j*. For the three datasets of 1 mm, 4 mm and 10 mm, based on the RAR*_i_*_, *j*_, the optimal parameter combinations (*I*^*^, *N*^*^, *k*^*^) were preferred, respectively, as follows:(9)$$\mathrm{RA}R_{i,j}\left(I^{*},N^{*},k^{*}\right)=\overset{}{\underset{\begin{array}{l} I\in \{780,782,\cdots ,2498\}\\ N\in \{2,3,\cdots ,859,860\}\\ k\in \{1,2,\cdots ,5\} \end{array}}{\mathrm{Max}}}\mathrm{RA}R_{i,j}(I,N,k),$$
(10)$$\mathrm{RA}R_{i,j}\left(I^{*},N^{*},k^{*}\right)=\overset{}{\underset{\begin{array}{l} I\in \{780,782,\cdots ,1882\}\\ N\in \{2,3,\cdots ,550,551\}\\ k\in \{1,2,\cdots ,5\} \end{array}}{\mathrm{Max}}}\mathrm{RA}R_{i,j}(I,N,k),$$
(11)$$\mathrm{RA}R_{i,j}\left(I^{*},N^{*},k^{*}\right)=\overset{}{\underset{\begin{array}{l} I\in \{780,782,\cdots ,1388\}\\ N\in \{2,3,\cdots ,304,305\}\\ k\in \{1,2,\cdots ,5\} \end{array}}{\mathrm{Max}}}\mathrm{RA}R_{i,j}(I,N,k).$$

The above algorithm was also written in MATLAB version 7.6 software.

Next, the three-model voting fusion was described. Taking Φ_1,2_, Φ_1,3_, Φ_2,3_ as sub-models for three-model voting: each sample was judged three times, then, the sample category of comprehensive judgment was determined according to the principle of “Two wins in three games”, so as to establish a joint model (Φ_Fusion_) of three-model voting.

Assuming that each compensation model (Φ_1,2_, Φ_1,3_, Φ_2,3_) has a high probability of identifying certain two categories of samples, then according to the principle of “two wins in three games” of three models voting, their joint model will likely have a high probability of identifying all three categories sample. Therefore, the total recognition accuracy rate (RAR_Total_) will likely reach a higher probability recognition effect, which is expected to be better than its three sub-models. See also Table 10.

In fact, the strategy proposed here can be generalized to the *n*-category discriminant analysis case: the *n* − 1 categories priority’s compensation models were first determined; the joint model based on *n*-model voting fusion was further established.

## 4. Conclusions

Inspired by aquaphotomics, a novel method of multi-modal spectra fusion modeling based on multi-optical path measurement was proposed. The optical path length of the transmission measurement accessory was regarded as a perturbation factor. The three-model voting fusion modeling approach based on multi-measurement modals (1 mm, 4 mm, and 10 mm) was used to establish the NIR spectral discriminant analysis models of three categories of drinking water.

The MW-kNN combined with Norris derivative filter was used to optimize the three-category discriminant analysis models for each spectral modal. Drawing on the idea of game theory, the selection method of the two-category priority compensation models was proposed, and the fusion modeling method of three compensation models’ voting was further proposed, aiming to make the joint models achieve a better comprehensive discriminant effect.

Based on the model set of MW-kNN with NDF, the global optimal models and the two-category priority compensation models for each modal were determined, and the joint models based on the fusion of single-modal or multi-modal models were also determined. The comprehensive discrimination effects of all joint models were better than their three sub-models; the effect of multi-modal fusion was better than that of single-modal fusion. The joint voting of the multi-modal compensation models can highlight the spectral differences from two dimensions, thereby improving the discrimination accuracy of the spectral population with small differences. The global optimal joint model was the dual-modal fusion of 1 mm-Φ_1,2_, 10 mm-Φ_1,3_, and 1 mm-Φ_2,3_, and the validation’s RAR_Total_ reached 95.5%. Thus, it showed the feasibility of NIR spectroscopy for the multi-classification identification of drinking water.

Furthermore, the MWCC spectrum, inter-category, and intra-category MWCC spectra were also proposed to evaluate small differences between two spectral populations of drinking water samples. Based on the principle of kNN and the wavelength model optimization, the plotting approach of the first *k*-shortest distances was also proposed to describe the differences in spectral populations. Their results showed the spectral differences between the three categories of water sample populations.

In summary, a novel NIR spectral pattern recognition strategy was proposed, which included two aspects: three compensation models’ voting and multi-modal fusion, which can significantly improve the discriminant analysis effect of spectral populations with small differences. The idea of this methodology is also expected to be widely applied to spectral discriminant analysis in other fields.

## Figures and Tables

**Figure 1 molecules-27-04485-f001:**
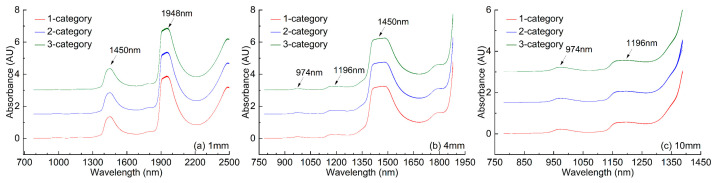
Spectra in desaturated wavebands of three groups of water samples based on three measurement modals: (**a**) 1 mm, (**b**) 4 mm, (**c**) 10 mm.

**Figure 2 molecules-27-04485-f002:**
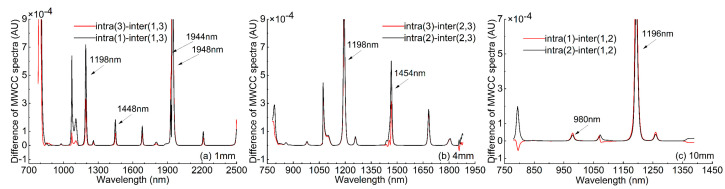
Difference spectra between intra-category and inter-category MWCC spectra: (**a**) 1 mm, 1–3 category, (**b**) 4 mm, 2–3 category, (**c**) 10 mm, 1–2 category.

**Figure 3 molecules-27-04485-f003:**
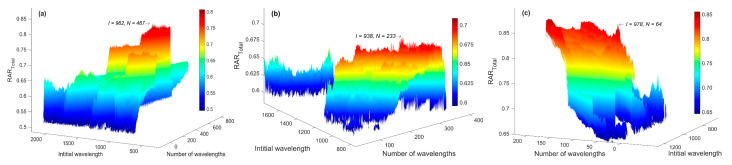
The 3D effect diagrams of the modeling discrimination effect of MW-kNN models of three measurement modals for the initial wavelength (*I*) and the number of wavelengths (*N*): (**a**) 1 mm, (**b**) 4 mm, (**c**) 10 mm.

**Figure 4 molecules-27-04485-f004:**
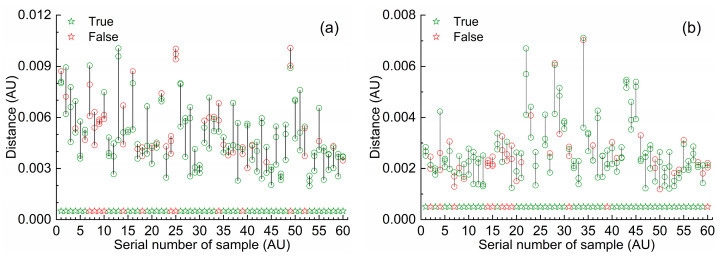
First three shortest distances between each prediction sample and all the calibration samples based on the optimal MW-kNN waveband in the three measurement modals: (**a**) 1-category, 1 mm, 966–1894 nm; (**b**) 2-category, 4 mm, 938–1402 nm; (**c**) 3-category, 10 mm, 964–1378 nm.

**Figure 5 molecules-27-04485-f005:**
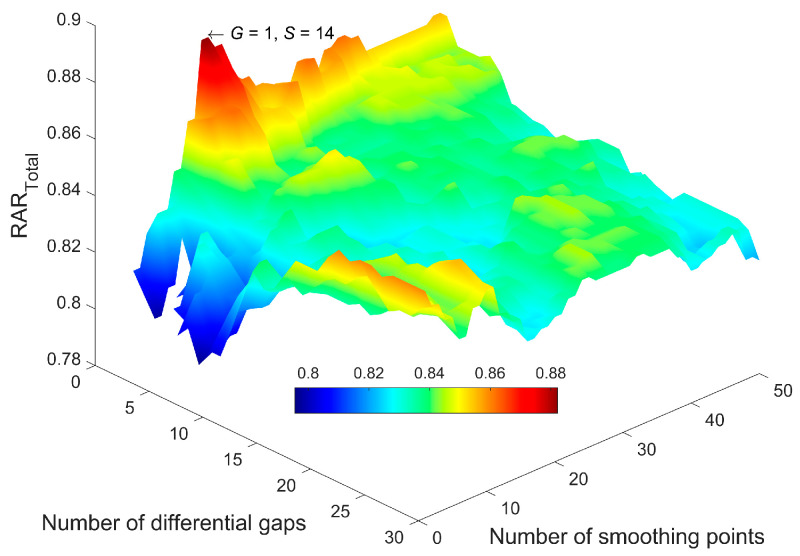
The 3D effect diagram of the modeling discrimination effect of kNN models processed by NDF for the number of smoothing points (*S*) and the number of differential gaps (*G*).

**Figure 6 molecules-27-04485-f006:**
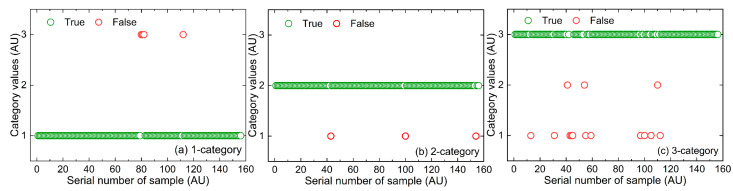
Schematic diagrams of the recognition effect of the optimal joint model based on three-model voting fusion for three categories of validation samples: (**a**) 1-category, (**b**) 2-category, (**c**) 3-category.

**Table 1 molecules-27-04485-t001:** Modeling discrimination effects of kNN models of three measurement modals based on desaturated wavebands.

Modal	Waveband (nm)	*N*	*k*	RAR_Total_
1 mm	780–2498	860	4	64.4%
4 mm	780–1880	551	1	44.9%
10 mm	780–1388	305	3	82.9%

**Table 2 molecules-27-04485-t002:** Modeling discrimination effects of the optimal MW-kNN models of three measurement modals.

Modal	*I*	*N*	*k*	RAR_Total_
1 mm	966	465	3	80.7%
4 mm	938	233	3	70.9%
10 mm	964	208	3	85.3%

**Table 3 molecules-27-04485-t003:** Modeling discrimination effects of the 2-category priority’s compensation models of three measurement modals.

Modal	Submodel	*I*	*N*	*k*	RAR_1,2_	RAR_1,3_	RAR_2,3_	RAR_Total_
1 mm	Φ_1,2_	1704	83	2	81.0%			68.0%
	Φ_1,3_	858	520	3		84.7%		78.9%
	Φ_2,3_	844	524	1			80.7%	76.9%
4 mm	Φ_1,2_	798	308	4	85.3%			69.6%
	Φ_1,3_	986	209	3		66.0%		70.0%
	Φ_2,3_	1080	127	1			72.3%	70.0%
10 mm	Φ_1,2_	970	205	4	78.3%			85.1%
	Φ_1,3_	960	88	4		95.3%		82.2%
	Φ_2,3_	924	231	3			90.3%	85.1%

**Table 4 molecules-27-04485-t004:** Modeling discrimination effects of the optimal MW-kNN models processed by optimal NDF parameters.

Modal	*I*	*N*	*k*	*D*	*S*	*G*	RAR_Total_
1 mm	966	465	3	1	14	1	88.2%
4 mm	938	233	3	2	2	1	73.8%
10 mm	964	208	3	2	1	1	85.6%

**Table 5 molecules-27-04485-t005:** Modeling discrimination effect of the 2-category priority’s compensation models of three measurement modals processed by optimal NDF parameters.

Modal	Submodel	*D*	*S*	*G*	*k*	RAR_1,2_	RAR_1,3_	RAR_2,3_	RAR_Total_
1 mm	Φ_1,2_	2	26	5	2	98.3%			88.2%
	Φ_1,3_	1	18	2	3		88.3%		85.6%
	Φ_2,3_	1	12	1	1			90.0%	88.9%
4 mm	Φ_1,2_	0	4	--	4	86.0%			69.8%
	Φ_1,3_	2	6	2	3		71.3%		73.3%
	Φ_2,3_	1	36	6	1			77.3%	73.6%
10 mm	Φ_1,2_	2	1	1	3	78.7%			85.6%
	Φ_1,3_	2	20	5	2		95.7%		77.8%
	Φ_2,3_	0	3	--	3			90.0%	85.1%

**Table 6 molecules-27-04485-t006:** Modeling discrimination effect of the joint models for three compensation models’ voting based on single-modal.

Modal	Submodel			RAR_1_	RAR_2_	RAR_3_	RAR_Total_	RAR_SD_
1 mm	Φ_1,2_	Φ_1,3_	Φ_2,3_	96.0%	94.0%	90.0%	93.3%	3.1%
4 mm	Φ_1,2_	Φ_1,3_	Φ_2,3_	76.0%	94.7%	68.7%	79.8%	13.4%
10 mm	Φ_1,2_	Φ_1,3_	Φ_2,3_	85.3%	75.3%	100.0%	86.9%	12.4%

**Table 7 molecules-27-04485-t007:** Modeling discrimination effects of the optimal joint models for three-model voting fusion based on multi-modal.

Modal	Modal (mm)-Submodel			RAR_1_	RAR_2_	RAR_3_	RAR_Total_	RAR_SD_
Dual modal	1-Φ_1,2_	10-Φ_1,3_	1-Φ_2,3_	98.7%	95.3%	94.0%	96.0%	2.4%
Triple-modal (global)	1-Φ_global_	4-Φ_global_	10-Φ_global_	89.3%	92.0%	98.7%	93.3%	4.8%
Triple-modal (compensatory)	1-Φ_1,2_	10-Φ_1,3_	4-Φ_2,3_	97.3%	96.7%	92.7%	95.6%	2.5%

**Table 8 molecules-27-04485-t008:** Comparison of the modeling discrimination effects for the optimal joint models of the three-model voting fusion and their sub-models.

Modal	Modal (mm)-Submodel			Joint Model RAR_Total_	Increase Rate of RAR_Total_		
	Model 1	Model 2	Model 3		Model 1	Model 2	Model 3
Single modal	1-Φ_1,2_	1-Φ_1,3_	1-Φ_2,3_	93.3%	5.1%	7.7%	4.4%
Dual modal	1-Φ_1,2_	10-Φ_1,3_	1-Φ_2,3_	96.0%	18.2%	7.8%	7.1%
Triple-modal (global)	1-Φ_global_	4-Φ_global_	10-Φ_global_	93.3%	5.1%	19.5%	7.7%
Triple-modal (compensatory)	1-Φ_1,2_	10-Φ_1,3_	4-Φ_2,3_	95.6%	7.4%	22.0%	17.8%

**Table 9 molecules-27-04485-t009:** Validation discrimination effect of the four optimal joint models based on the fusion of single-modal or multi-modal models.

Modal	RAR_1_	RAR_2_	RAR_3_	RAR_Total_	RAR_SD_
Single modal	95.5%	94.2%	84.0%	91.2%	6.3%
Dual modal	97.4%	98.1%	91.0%	95.5%	3.9%
Triple-modal (global)	88.5%	91.7%	98.1%	92.7%	4.9%
Triple-modal (compensatory)	95.5%	99.4%	90.4%	95.1%	4.5%

**Table 10 molecules-27-04485-t010:** Schematic diagram of voting effect for the three-model fusion’s joint model and its sub-models.

Model	1-Category	2-Category	3-Category	All Samples
Φ_1,2_	High	High		High
Φ_1,3_	High		High	High
Φ_2,3_		High	High	High
Φ_Fusion_	High	High	High	Higher

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
