# Peer review of "Performance Improvement of NIR Spectral Pattern Recognition from Three Compensation Models’ Voting and Multi-Modal Fusion"

_molecules, 2022, doi:10.3390/molecules27144485_

Round 1

Reviewer 1 Report

Main remarks

The manuscript presents an interesting study related to the possibility of authenticity identification of drinking water brands.

In general, there is no explanation of the spectral basis for distinguishing water brands. A better description of the methods used to process the spectral characteristics of the analyzed samples is also needed.

Additional remarks

Section 2.2 – “Intra- and inter-category MWCC spectra of spectral populations of any two categories of samples were calculated to evaluate the weak differences between the two categories of water spectra”. Figure 2 shows the difference spectra, but there is no analysis of the results - what water structures can be associated with the differences.

Line 257-258 – “the green spots were generally lower than the red spots” – it's not clear.

Line 356-357 – “The three modals were arranged sequentially, and there were six cases as follows: 1-4-10, 1-10-4, 4-1-10, 4-10-1, 10-1-4, 10-4-1” - Is there a difference in the accuracy of models for the different arranged groups, since the spectral information is the same?

Figure 6  is not very clear, some of the points are difficult to see.

In the section "Material and methods" the description of some items must be supplemented:

-          - What software is used for NDF and MW-kNN?

-          - Explanation of the selection method of the 2-category priority compensation models.

Reviewer 2 Report

Dear Editor

This paper is discussed the analysis of three categories of drinking water using Near-infrared (NIR) spectroscopy with multi-measurement modals, depending on the idea of Game Theory. The writing style of paper is very good and the author(s) explain the practical work and the results clearly and high precise. The paper acquires its important from the global continuous decreases of drinking water sources. Therefore, this paper is so suitable to publish in your respected journal.

Author Response

Thank you very much for your comments.

We have made comprehensive and detailed revisions.

Round 2

Reviewer 1 Report

The manuscript has been significantly improved.